

# Detection of offensive content in the Kazakh language using machine learning and deep learning approaches

Milana Bolatbek, Moldir Sagynay, Shynar Mussiraliyeva and Zhastay Yeltay

Department of Cybersecurity and Cryptology, Al-Farabi Kazakh National University, Almaty, Kazakhstan

## ABSTRACT

This article addresses the urgent need to detect destructive content, including religious extremism, racism, cyberbullying, and nation oriented extremism messages, on social media platforms in the Kazakh language. Given the agglutinative structure and rich morphology of Kazakh, standard natural language processing (NLP) models require significant adaptation. The study employs a range of machine learning and deep learning techniques, such as logistic regression, support vector machines (SVM), and long short-term memory (LSTM) networks, to classify destructive content. This article demonstrates the effectiveness of combining n-gram and stemming methods with machine learning algorithms, achieving high accuracy in content classification. The findings underscore the importance of developing language-specific NLP tools tailored to Kazakh's linguistic complexities. This research not only contributes to ensuring online safety by detecting destructive content in Kazakh digital spaces, but also provides a framework for applying similar techniques to other lesser-resourced languages.

## INTRODUCTION

The digital transformation of the 21st century has dramatically altered how individuals and communities interact, giving rise to new opportunities for social engagement as well as new challenges (*Suherlan, 2023*). One of the most pressing issues in this digital landscape is the spread of destructive speech, including hate speech, cyberbullying, and extremist content (*Dreißigacker et al., 2024*; *Perera et al., 2023*; *Jahan & Oussalah, 2023*; *Fulantelli et al., 2022*; *Castano-Pulgarin et al., 2021*).

The Internet is actively used to spread harmful and destructive messages (*Iskhakova, Iskhakov & Meshcheryakov, 2019*). The Internet provides a convenient platform for such activities, and social networks, blogs, forums, and other resources allow users to express their opinions anonymously (*Okhapkina et al., 2020*). These platforms also facilitate recruiting new members, spreading false information, misleading the public, and distributing material that infringes intellectual property rights. Through social networks, extremist organizations can find like-minded people and attract them to their ranks.

Corresponding authors
Milana Bolatbek,
bolatbek.milana@gmail.com
Moldir Sagynay,
sagynaymoldir11@gmail.com

Young people are particularly susceptible to the influence of extremist groups, as they are actively influenced by destructive messages spread through the Internet and digital communications (*Shmelev et al., 2020*). These forms of harmful communication not only threaten individual safety and mental health but also pose significant risks to social cohesion and national security (*Haghish et al., 2023*).

Despite significant advances in destructive content detection, existing research has primarily focused on high-resource languages (*Sharif et al., 2024*; *Rawat, Kumar & Samant, 2024*; *Parihar, Thapa & Mishra, 2021*; *Mnassri et al., 2024*; *Al-Dabet et al., 2023*; *Berjawi, Fenza & Loia, 2023*; *Gaikwad et al., 2023*; *Berhoum et al., 2023*; *Gaikwad et al., 2022*; *Sigurbergsson & Derczynski, 2019*). Studies addressing low-resource languages like Kazakh remain extremely limited. Most previous approaches either rely on traditional machine learning techniques using handcrafted features or apply transformer-based models without tailoring them to the linguistic characteristics of specific languages. These methods often fail to capture the complex agglutinative morphology and sequential dependencies present in Kazakh texts, which can lead to reduced classification performance. To address these limitations, this study proposes a hybrid deep learning model that combines pre-trained Bidirectional Encoder Representations from Transformers (BERT) embeddings with an long-short term memory (LSTM) layer. By doing so, the model not only benefits from rich contextual word representations but also enhances sequential modeling, allowing it to better capture the linguistic structure of Kazakh, by evaluating datasets specifically for Kazakh destructive speech detection, the study directly contributes to addressing the lack of language-specific resources in this domain.

Kazakhstan, a Central Asian country with a population of over 19 million (*Bureau of National Statistics of the Republic of Kazakhstan, 2023*), is characterized by its diverse ethnic and linguistic composition. According to a 2023 survey, at the beginning of 2023, Kazakhstan had 17.73 million internet users, with an internet penetration rate of 90.9%. In January 2023, the country had 11.85 million social media users, representing 60.8% of the total population. Additionally, there were 25.44 million active cellular mobile connections in Kazakhstan, which amounted to 130.5% of the population (*Datareportal, 2023*).

However, the online environment in Kazakhstan is increasingly being exploited for the dissemination of destructive speech, including ethnic slurs, religious extremism, and gender-based violence. In 2022, a national study conducted by the National Center for Public Health of Kazakhstan's Ministry of Health revealed that 17.5% of children in Kazakhstan experience occasional bullying. The survey also found that 6.8% of teenagers face bullying and intimidation 2–3 times a month, while 14.1% of children aged 11–15 admitted to participating in bullying their peers (*Ministry of Health of the Republic of Kazakhstan, 2023*).

The detection of destructive speech in Kazakh is a complex task due to several linguistic and technical challenges. The Kazakh language, with its agglutinative structure and rich morphology, requires sophisticated natural language processing (NLP) models capable of understanding context, inflection, and word formation. Kazakh exhibits significant dialectal variation, influenced by regional differences. This linguistic diversity complicates

the development of generalized models for speech detection. Furthermore, the digital infrastructure for Kazakh is relatively underdeveloped. Unlike English, which benefits from vast datasets and advanced AI models, Kazakh lacks large-scale annotated corpora and pre-trained language models. For example, as of 2023, there are only a handful of publicly available Kazakh language datasets, most of which are small and lack the diversity needed for robust model training.

Addressing these challenges requires a multi-faceted approach that combines technological innovation with cultural and ethical considerations. This article aims to examine existing machine learning and deep learning models in detecting destructive content in Kazakh.

As Kazakhstan continues its digital evolution, ensuring the safety and well-being of its online communities is paramount. Developing effective methods for detecting destructive speech in the Kazakh language is not only a matter of national security but also a crucial step in preserving the cultural integrity of the Kazakh people. This article seeks to contribute to the ongoing dialogue on responsible AI development, highlighting the importance of inclusive and culturally aware approaches to NLP in the context of lesser-known languages.

Considering the above conclusions, it can be concluded that the task of creating models for identifying destructive messages for the classification of destructive web content in the Kazakh language is the most urgent. The primary contributions of this study are as follows:

(1) a systematic evaluation of traditional machine learning models based on Term Frequency-Inverse Document Frequency (TF-IDF) and n-gram features for the classification of destructive content;

(2) the proposal and assessment of a hybrid deep learning model combining BERT embeddings with an LSTM layer to better capture sequential dependencies in morphologically rich Kazakh texts;

(3) a critical comparison between traditional and deep learning approaches to highlight the challenges and advantages of language-specific modeling for destructive content detection.

The novelty of the proposed method lies in its adaptation of a hybrid architecture combining BERT embeddings with an LSTM layer, specifically designed for destructive speech detection in the Kazakh language. While BERT provides strong contextual representations, it does not fully capture sequential dependencies that are particularly important for agglutinative languages like Kazakh. By integrating an LSTM layer after the BERT embeddings, the proposed model addresses the linguistic complexity of Kazakh more effectively than models relying solely on transformer outputs. Furthermore, unlike previous studies that focus predominantly on high-resource languages such as English, our approach targets a low-resource linguistic environment, using a newly developed and annotated dataset. This dual focus on hybrid architecture and low-resource language application differentiates our work from existing studies and advances research on destructive content detection in underrepresented languages.

This article begins with an Introduction to the significance of the issue and the challenges posed by Kazakh's unique linguistic features. "Literature Review" describes related work in the field of destructive speech detection. "Materials and Methods" details the approaches used to develop and test detection models, including data collection and preprocessing. Experimental results are presented, showcasing the performance of these models. The article concludes with a summary of findings and suggestions for future research, followed by a comprehensive list of references.

## Defining destructive speech

Destructive speech encompasses various forms of communication that cause harm or promote negative outcomes, particularly within digital environments. The categories of bullying, racism, nation-based extremism, and religious extremism were selected for this study because they represent the most prevalent and socially significant forms of destructive speech identified in the Kazakh online environment. Prior studies conducted by author's research group revealed that these categories are particularly common in Kazakh-language hate speech, making them critical targets for classification. Each of which poses unique challenges and risks in the digital context:

1. Religious extremism refers to the ideology held by certain movements, groups, or individuals within religious denominations and organizations, characterized by a strict adherence to extreme interpretations of religious doctrine (*Eraliev, 2022*).
2. Cyberbullying is an act of using the Internet to inflict harm or fear on another person, particularly by sending them distressing or threatening messages, known as online harassment (*Cambridge Dictionary, 2024*).
3. National origin discrimination (national extremism) occurs when individuals are treated unfairly due to their country of origin, ethnicity, accent, or perceived ethnic background, regardless of whether these characteristics are accurate (*U.S. Equal Employment Opportunity Commission, 2024*).
4. Racism involves discrimination and bias directed at individuals due to their race or ethnicity (*Wikipedia Contributors, 2024*).

## Motivation

The rapid advancement of digital technologies has significantly transformed communication patterns across the globe, offering unprecedented opportunities for interaction and information exchange. However, this digital revolution has also introduced new challenges, particularly in the realm of online destructive speech. In Kazakhstan, where the Kazakh language is a cornerstone of national identity and cultural heritage, addressing the issue of harmful online content becomes crucial for maintaining social harmony and protecting individuals from harm.

Despite the growing presence of the Kazakh language in digital spaces, there is a notable scarcity of effective tools and methodologies for detecting destructive speech in Kazakh. The unique linguistic characteristics of Kazakh, including its agglutinative structure, dialectal diversity, and substantial influence from other languages, present significant

hurdles for conventional natural language processing (NLP) techniques. These challenges are compounded by the limited availability of annotated datasets and language-specific resources, which are essential for training accurate and reliable detection models.

The motivation for this article stems from the urgent need to bridge this gap and develop effective solutions for detecting destructive speech in the Kazakh language. By addressing the specific linguistic and contextual challenges, this research aims to contribute to the broader field of NLP and artificial intelligence (AI), providing tools that can enhance the safety and well-being of Kazakh-speaking online communities. This study seeks to promote responsible AI development by ensuring that detection systems are culturally sensitive and aligned with ethical standards.

Through this research, we aspire to advance the understanding of destructive speech detection in less commonly spoken languages and to offer practical solutions that can be applied in Kazakhstan and similar contexts. The ultimate goal is to create a safer digital environment where individuals can engage in meaningful dialogue without fear of harm or discrimination, thus supporting the principles of free expression while mitigating the risks associated with online destructive speech.

## LITERATURE REVIEW

The study of destructive speech detection has garnered significant attention in recent years, with various approaches developed to tackle this issue across different languages. Existing research often focuses on hate speech and cyberbullying detection, employing machine learning and natural language processing techniques to identify harmful content. Addressing this issue is crucial, as failure to do so can lead to serious negative consequences. However, most of these studies primarily address English and other widely spoken languages, leaving a gap in the application of these techniques to less commonly studied languages such as Kazakh. For instance, *Arbaatun, Nurjanah & Nurrahmi (2022)* utilized Twitter posts and applied an LSTM model to detect hate speech. Another study— *Aliyeva & Yağanoğlu (2024)*—attempted to identify cyberbullying in Turkish. However, these studies did not account for the grammatical complexities of agglutinative languages.

One line of research introduced a General Risk Index derived from psychological indicators in user-generated text. Validated through receiver operating characteristic (ROC) analysis, the model achieved a high classification accuracy of 90–96% in distinguishing individuals with violent tendencies from general users (*Kaati, Shrestha & Akrami, 2023*). While this demonstrates the potential of large-scale textual analysis as an early warning tool, the study was conducted exclusively in English, leaving its applicability to morphologically rich or low-resource languages largely unexplored.

One notable contribution in the field of violent extremism detection is the work of (*Abd-Elaal, Badr & Mahdi, 2020*), who proposed an intelligent system for detecting Pro-ISIS accounts on Twitter by leveraging both linguistic and behavioral features. The study introduces a dual-subsystem architecture—crawling and inquiring—that autonomously tracks and assesses user accounts.

Another significant contribution to the domain of online extremism detection is the study analyzing the Proud Boys movement, a contemporary radical extremist group that

has effectively leveraged social media platforms to disseminate its ideological narratives. The authors of this research conducted a multi-faceted analysis of social media discourse surrounding Proud Boys-related protests, focusing on (i) user profiles and ideological leanings, (ii) network structures and community formations, and (iii) tweet-level engagement metrics. The findings reveal that support for Proud Boys predominantly originates from conservative, religious, and right-wing segments who justify their stance through appeals to patriotism and American values (*Nguyen & Gokhale, 2022*).

## Cyberbullying detection using machine learning models

The application of machine learning techniques for detecting extremist content, hate speech, and cyberbullying has seen considerable advancement, particularly in high-resource languages.

A notable contribution in this direction is presented in *Bolatbek & Mussiraliyeva (2023)*, where the authors develop semantic analysis models trained specifically on Kazakh-language data to identify extremist content. Utilizing bigrams and word embedding approaches, the study reports strong performance across multiple machine learning classifiers. By constructing and testing models on a dedicated Kazakh-language *corpus*, this research not only contributes a practical framework for local language content moderation but also marks a critical step forward in low-resource language adaptation.

*Heidari, James & Uzuner (2021)* offers a comparative assessment of several machine learning algorithms—including support vector machine, random forests, logistic regression, and neural networks—for the task of bot account detection. While these methods demonstrate high accuracy and speed, their applicability remains constrained by monolingual training data. The absence of cross-linguistic evaluation limits the transferability of the findings to languages with different syntactic and semantic characteristics.

A systematic review presented in *Mansur, Omar & Tiun (2023)* explores a taxonomy of hate speech detection approaches, ranging from traditional rule-based methods to deep learning architectures such as transformers. Although the review provides valuable insights into algorithmic performance and key challenges such as contextual ambiguity and evolving hate speech patterns, the evaluation is again limited to English datasets. As a result, the review overlooks the performance variance that may arise in multilingual or morphologically complex environments.

Similarly, the study in *Ayo et al. (2020)* compares various deep learning approaches—including recurrent neural networks (RNNs), transformers, and support vector machines (SVMs)—on hate speech classification tasks.

Recent contributions such as *Yadav, Bajaj & Gupta (2021)* further investigate the use of neural networks, hybrid models, and contextual embeddings in detecting offensive and hateful speech. These studies highlight technical challenges including imbalanced datasets and ambiguous terms. However, their models are trained exclusively on English-language content, raising concerns about the effectiveness of such systems in other linguistic contexts.

An innovative approach explored in *Aljero & Dimililer (2021)* employs genetic programming to automatically generate classifiers for hate speech detection. The method is praised for its adaptability and accuracy, but its evaluation also remains confined to English data, leaving questions about its cross-lingual generalization potential.

A critical dimension in the development of hate speech recognizers (HSRs) lies in addressing unintended bias and ethical concerns in automated content moderation. In this context, a recent study introduced KERM-HATE, a syntax-based HSR framework designed to reduce prejudice effects often observed in state-of-the-art models. KERM-HATE leverages syntax heat parse trees as post-hoc explanations to enhance model interpretability and emphasize syntactic patterns over potentially biased semantic cues. The system outperformed established architectures such as BERT, RoBERTa, and XLNet on standard benchmark datasets, demonstrating superior classification performance (*Mastromattei et al., 2022*).

### Racism detection using machine learning models

The study in *Vidgen & Yasseri (2020)* explores methods for detecting both mild and explicit forms of Islamophobic hate speech on social media. Using natural language processing and machine learning techniques, the authors develop models capable of distinguishing varying levels of hostility. The importance of contextual interpretation is emphasized, with deep learning approaches enhancing detection accuracy.

### National origin extremism detection using machine learning models

Previous work by the authors of this article has significantly contributed to the field of extremism detection, with several publications addressing various aspects of identifying extremist content in Kazakh language (*Mussiraliyeva et al., 2023*, *2021*; *Zhenisbekovna, Aslanbekkyzy & Bolatkyzy, 2024*; *Bolatbek et al., 2024*).

In the majority of reviewed studies, model development and evaluation rely heavily on English-language datasets. Although multilingualism is often acknowledged at a theoretical level, practical adaptation and cross-linguistic validation of models are rarely implemented. Preprocessing and modeling techniques tailored to the morphological characteristics of languages like Kazakh are still scarce.

Our article seeks to address this gap. Multiple machine learning models were trained and evaluated, resulting in the first comparative analysis of destructive content detection methods specifically tailored to Kazakh. This research highlights the necessity and feasibility of language-specific approaches and represents an important step toward the development of inclusive NLP systems.

Table 1 presents a comparative analysis of recent studies employing large language models and traditional machine learning (ML) methods for hate speech detection. The studies span a range of languages and platforms—including English, Kazakh, Arabic, Urdu, and Hindi—and utilize widely adopted models such as BERT, RoBERTa, GPT-3, LSTM, support vector machines (SVM), decision trees, and genetic programming.

Although high accuracy and F1-scores have been reported, persistent challenges remain in the areas of language adaptation, contextual understanding, fairness, and

**Table 1 Comparative analysis of recent studies on hate speech detection using large language models and machine learning techniques.**

| Reference | Datasets | Key findings | Methodology and aproach | Scope | Limitation for Kazakh language |
|---|---|---|---|---|---|
| *Abd-Elaal, Badr & Mahdi (2020)* | Text data from 208,288 users across 32 online platforms, including 76 known lone violent offenders | The General Risk Index, based on textual indicators, achieved classification accuracy between 90–96% in detecting potential violent individuals | Automated text analysis; dictionary-based and ML methods to extract psychological risk indicators; ROC analysis | Risk and threat assessment through written communication on digital platforms | There are no psycholinguistic dictionaries or annotated datasets available for the Kazakh language. Since the model is based on English-language indicators and classifiers, it cannot be directly applied to Kazakh due to morphological and cultural differences. |
| *Abd-Elaal, Badr & Mahdi (2020)* | 21,000 Pro-ISIS tweets, 21,000 Anti-ISIS tweets, 21,000 Non-ISIS tweets (Arabic); Translated English datasets used | Achieved 89% F1-score for tweet-level detection and 94% F1-score for account-level detection using supervised learning models | Linguistic and behavioral feature extraction; TF-IDF, Skip-gram, Mazajak embeddings; classifiers: SVM, Naive Bayes, random forest | Detection of radicalization and classification of ISIS-related accounts on Twitter | The applied model, ISIS-Account Detector, is trained on English and Arabic datasets focusing on ISIS-related content. It lacks adaptation for the Kazakh language, which presents challenges due to its agglutinative morphology and limited linguistic resources. Consequently, the model cannot be directly applied to Kazakh without significant modifications. |
| *Bolatbek & Mussiraliyeva (2023)* | Kazakh-language extremist and neutral texts *corpus* (1,200 extremist messages, ~140,000 words). Data collected from VKontakte and open web sources. | The proposed TF-IDF _bigram _LSTM model achieved the highest performance (Accuracy: 0.90, F1-score: 0.88, AUC-ROC: 0.89) compared to classical ML models. | *Corpus* construction, annotation (1/0 labels), preprocessing (tokenization, stopword removal, stemming), feature extraction (TF-IDF, bigrams), deep learning (LSTM) and ML baselines (SVM, RF, NB). | Detection of extremist messages in the Kazakh language on social media using semantic analysis and machine learning techniques. | Although the study is focused on the Kazakh language, it highlights the lack of large-scale annotated datasets and standardized linguistic resources, which limits the performance and generalizability of extremist content detection models. |
| *Heidari, James & Uzuner (2021)* | Twitter bot dataset from Botometer (labeled accounts: bots and humans) | Random forest classifier achieved the highest accuracy (above 95%) in detecting bots. Deep learning underperformed in small datasets. | Empirical comparison of multiple ML classifiers: RF, SVM, DT, and DNN. Evaluation metrics: Accuracy, Precision, Recall, F1-score. | Detection and classification of social media bots on Twitter | The transformer models used in this study are trained on English data. Due to the lack of Kazakh-language training data and linguistic differences, the models cannot be directly applied to Kazakh without significant adaptation. |
| *Mansur, Omar & Tiun (2023)* | Twitter | Deep learning, especially transformer-based models, significantly outperform classical ML; challenges remain with multilingual and contextual hate. | Comprehensive review of deep learning models taxonomy of approaches; evaluation criteria discussed | Global overview of deep learning techniques in hate speech detection | Although the study focuses on low-resource languages, Kazakh was not included in the evaluation. The lack of labeled datasets and language-specific resources for Kazakh limits the applicability and performance of the multilingual models. |

| Reference | Datasets | Key findings | Methodology and aproach | Scope | Limitation for Kazakh language |
|---|---|---|---|---|---|
| *Ayo et al. (2020)* | Various public datasets from Twitter, Facebook, YouTube (*e.g.,* Hatebase, Waseem & Hovy, Kaggle hate speech data) | ML methods like SVM, RF, and ensemble models perform well; however, the challenge remains in domain transfer, sarcasm, and context understanding. | Systematic survey of classical ML models (SVM, DT, NB), feature engineering (TF-IDF, word embeddings), evaluation metrics overview | Broad analysis of anti-social and hate speech detection on social media | The models were trained on English tweets and do not account for Kazakh linguistic features. Due to the absence of annotated Kazakh data and agglutinative morphology, these deep learning models cannot be directly applied to Kazakh without major adaptation. |
| *Aljero & Dimililer (2021)* | Twitter dataset with hate and non-hate labeled content | Genetic programming classifiers achieved higher accuracy and precision than traditional ML models (like SVM, NB, RF) on the same dataset. | Applied genetic programming for evolving classifiers; feature extraction from tweets; compared with classical algorithms | Hate speech detection on social media using evolutionary computation methods | The proposed GP model is trained on English Twitter data using Universal Sentence Encoder. Due to the lack of Kazakh-language embeddings, annotated datasets, and the agglutinative structure of Kazakh, the model requires extensive adaptation for effective use in Kazakh. |
| *Mastromattei et al. (2022)* | Hate speech datasets | Syntactic features introduce unintended bias; models relying heavily on syntax tend to misclassify due to lack of contextual understanding. | Comparative analysis of syntactic *vs.* semantic features; tested ML models using different feature sets (POS, n-gram, embedding) | Analyzing algorithmic and linguistic bias in hate speech detection models | The KERM-HATE model relies on English syntax structures and was trained on ethically-biased English datasets. Due to the lack of syntactic parsers, annotated hate speech corpora, and culturally adapted evaluation tools for Kazakh, applying this model to the Kazakh language remains highly limited. |
| *Vidgen & Yasseri (2020)* | 140 min English-language tweets manually annotated for weak/ strong Islamophobic hate speech | Classifier performed well distinguishing between non-hateful, weakly Islamophobic, and strongly Islamophobic content (F1-score ≈ 0.80). | Manual annotation, SVM-based classification; linguistic and thematic feature engineering; 3-class categorization | Detection and categorization of Islamophobic hate speech on Twitter | The model is trained on English-language data and tailored to Western cultural contexts. Due to the lack of annotated Kazakh datasets, differences in linguistic structure, and distinct cultural expressions of Islamophobia, the model requires significant adaptation for effective application to the Kazakh language. |

interpretability—especially for low-resource languages such as Kazakh and Arabic. This review clearly underscores the need for adaptive, interpretable, and equitable systems to address hate speech effectively in a multilingual digital landscape.

# MATERIALS AND METHODS

This section outlines the approach used to develop and evaluate models for detecting destructive speech in the Kazakh language. This section details the data collection, preprocessing, model selection, training, and evaluation processes (Fig. 1).

The general methodology for identifying destructive messages is shown below: the first stage is to collect data, that is, to create a *corpus* from the collected data (opinions). The second stage is the preliminary processing of collected *corpus* texts. Algorithms of tokenization, removal of stop words, stemming, cleaning of unnecessary symbols are

performed. Natural language processing (NLP) techniques are used. The third stage is the formation of features, the characteristics of the text that can be used to identify destructive messages are determined. This includes using Uni-bi-gram, identifying keywords and phrases, as well as other textual features.

As shown in Fig. 1, the illustrated diagram reflects the actual experimental setup implemented in this research. Each stage—from preprocessing and stemming to vectorization and model training—was executed using the specified techniques and architectures. This end-to-end pipeline served as the operational basis for all model comparisons described in "Experimental Results".

## Data collection

In this study, authors used their own previously collected and published dataset of Kazakh-language texts, focusing on destructive speech. The dataset was compiled using public Application Programming Interfaces (APIs) from platforms such as YouTube, Telegram, and VK (*Bolatbek et al., 2024*). To extract relevant destructive speech content, a keyword-based crawling strategy was implemented. The list of keywords and phrases was systematically developed by analyzing a preliminary set of open-source destructive messages in the Kazakh language. This initial analysis helped identify common linguistic patterns, slurs, and terminology associated with cyberbullying, racism, religious extremism, and nationalistic extremism. Examples of selected keywords include "қорлау" (insult), "дискриминация" (discrimination), "радикализм" (radicalism), "ұлтшылдық" (nationalism). Additionally, typical hashtags and phrases frequently appearing in hate speech and extremist discourse were incorporated to enhance data retrieval effectiveness.

After filtering and cleaning over 100,000 raw entries, the final dataset included 10,190 annotated texts across five categories: cyberbullying (2,136 texts), racism (2,205 texts), religious extremism (2,110 texts), national extremism (2,076 texts), and neutral content (1,663 texts). This dataset was originally introduced at Table 2. Figure 2 demonstrates distribution of text length for corpora texts.

The annotation was performed manually by five trained native Kazakh speakers. All annotators held academic degrees in computer linguistics, computer science. Before beginning the annotation, a detailed set of guidelines was provided. These guidelines clearly defined each category (cyberbullying, racism, religious extremism, national extremism, neutral content), with examples for each class to ensure consistency. The annotation process followed these steps:

1. Annotators independently labeled each text according to the provided category definitions.

2. Regular cross-check sessions were held to discuss ambiguous cases and resolve disagreements.

3. Annotated data was reviewed collectively to ensure consistency across annotators.

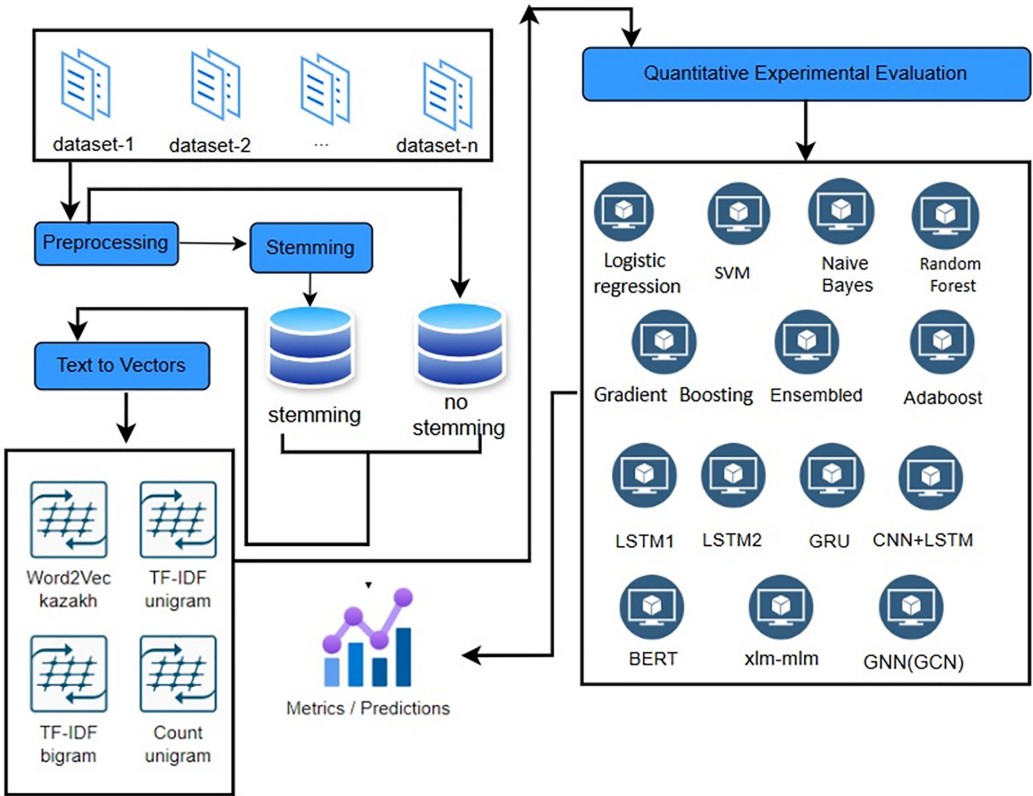

**Figure 1** Experimental setup for destructive speech detection in Kazakh.

To measure annotation reliability, Cohen's kappa coefficient was calculated between annotators. The resulting agreement score was 0.82, indicating a substantial level of agreement and confirming the consistency of the annotations.

The dataset collection and annotation processes strictly adhered to ethical standards. Only publicly available posts and comments were used, and no personal user data or sensitive private information was collected. Annotators were instructed to avoid any personally identifiable information. Given the use of publicly accessible data, and the focus on textual content rather than individuals, formal ethical approval was not required according to the applicable research guidelines. Nevertheless, all procedures were conducted in alignment with responsible research practices to ensure respect for user privacy.

## Data preprocessing

The collected text data underwent several preprocessing steps to prepare it for analysis:

- Text Cleaning. Removal of irrelevant information, such as HTML tags, special characters, and excessive whitespace.
- Tokenization. Splitting text into words or subwords using a tokenization algorithm tailored for the Kazakh language.

- Normalization. Converting text to a consistent format, including lowercasing and handling of diacritics.
- Stopword Removal. Eliminating common words that do not contribute to the meaning of the text.
- Stemming. Stemming was applied to reduce morphological variations in the Kazakh language, facilitating better matching of word forms while avoiding the complexity and potential data sparsity issues introduced by full lemmatization. More advanced text representation techniques, such as word embeddings (*e.g.*, Word2Vec, FastText), typically require substantially larger datasets to achieve optimal performance. Given the size of the current dataset and the focus on interpretability and computational efficiency, stemming was deemed the most appropriate choice for the traditional machine learning baseline models.
- The dataset was split into training (70%), validation (15%), and testing (15%) sets using stratified random sampling to ensure class balance across splits.

These preprocessing steps were selected to enhance the quality of textual features, reduce noise in the data, and ensure that linguistic patterns specific to the Kazakh language could be effectively captured. They provide a balanced trade-off between preserving semantic meaning and maintaining computational efficiency necessary for building robust machine learning models.

### Feature extraction and machine learning models

Feature selection is a crucial step in developing effective models for detecting destructive speech. In this study, we employed two primary techniques for feature extraction: TF-IDF and n-grams. This choice was based on extensive prior research demonstrating that TF-IDF is effective in capturing important lexical features for text classification tasks, especially in low-resource settings where deep learning models may underperform due to limited data availability *Aljwari et al. (2022)*.

TF-IDF is a statistical measure used to evaluate the importance of a word in a document relative to a collection of documents (*corpus*). It combines two factors: term frequency (TF), which measures how often a word appears in a document, and inverse document frequency (IDF), which assesses how unique or rare a word is across the entire *corpus*.

Term frequency (TF): measures how often a word appears in a document.

$$TF(w, d) = \frac{\text{Number of times word appears in document } d}{\text{Total number of words in document } d} \tag{1}$$

where $w$ is the word, and $d$ is the document.

Inverse document frequency (IDF): measures how important a word is across the entire *corpus*.

$$IDF(w, D) = log\left(\frac{\text{Total number of documents in corpus } D}{\text{Number of documents containing word } w}\right) \tag{2}$$

where $w$ is the word, and $D$ is the *corpus* of documents.

**Table 2** Dataset size and composition.

| | Label | Message | English |
|---|---|---|---|
| 0 | Violent | біздің сарбаздарымыз өз істерінің әділдігімен қаруланып басқыншы армия арасында көптеген шығындарға ұшырады және өз жерлері мен халқын қорғау үшін күресетін болады | Our soldiers, armed with the righteousness of their cause, have suffered many losses among the invading army and will fight to protect their land and people |
| 2 | Violent | менің халқым жалғыз емес бостандықты сүйетін миллиондаған ерлер мен әйелдер оның әділеттілік пен азаттық үшін күресуінде тұр | My people are not alone millions of freedom loving men and women are in his fight for justice and freedom stand |
| … | … | … | … |
| 391 | Neutral | адамның басшысы ақыл | The head of man is mind |
| … | … | … | … |
| 3028 | Bullying | тыңдашы аға мынадай сөздерді айтатын болсаң сен ауру шығарсың | Listen, brother, if you say the following words, you must be sick |
| 3029 | Bullying | қамап не керегі бар атып тастау керек | Should be locked up or shot |
| … | … | … | … |
| 4297 | Racism | мен негрлерімді сағынатын боламын | I will miss my niggers |
| 4299 | Racism | дұрыс маған негр керек | Right i need a nigger |
| … | … | … | … |
| 10187 | Nazism | онда жұмыс орындары бар бірақ жұмысқа рұқсат алу қымбатқа түседі шетелдіктердің көпшілігі қағазбастылықты болдырмау үшін бизнеспен айналысады | There are jobs there but getting a work permit is expensive and most foreigners do business to avoid paperwork |
| 10189 | Nazism | ұлттық қауіпсіздікке заңсыз иммигранттар қауіп төндіреді бірақ бұл ұлттық құзырет | National security is threatened by illegal immigrants, but that is a national competence |

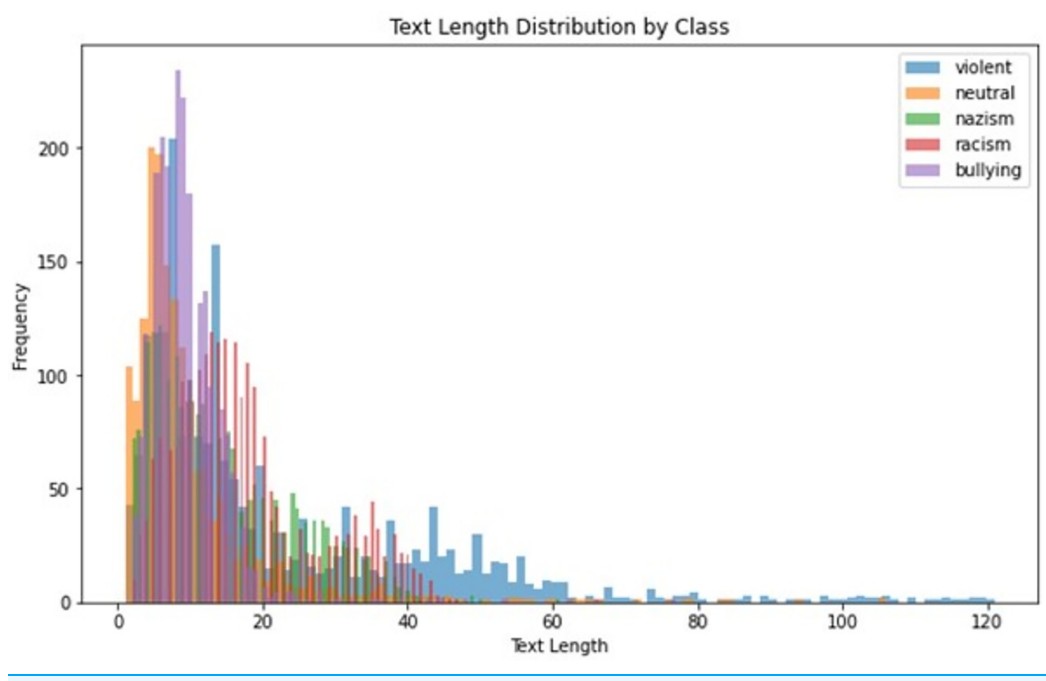

**Figure 2** Distribution of text length for the text.     

TF-IDF: Combines TF and IDF to give a balanced measure of a word's importance in a document relative to the *corpus*.

$$TF\text{-}IDF(w, d, D) = TF(w, d) \times IDF(w, D). \tag{3}$$

We applied TF-IDF to transform the text data into a numerical format that reflects the significance of each term within the context of destructive speech detection. This method helps to highlight terms that are more relevant to the classification task while reducing the impact of common words that occur frequently across documents.

*N*-grams are contiguous sequences of *n* words (or tokens) extracted from the text. They capture contextual relationships and patterns within the text that single words alone may not convey. For instance, bigrams (2-grams) and trigrams (3-grams) help to understand phrases or combinations of words that frequently occur together. We generated *n*-grams from the preprocessed text data, including both bigrams and trigrams, to capture more nuanced patterns in destructive speech. By incorporating these multi-word features, the model can better recognize context-specific expressions of harmful content.

$$n\text{-}gram = w_i, w_{(i+1)}, w_{(i+2)}, \ldots, w_{(i+n-1)} \tag{4}$$

where $i$ ranges from 1 to $T - n + 1$.

*Feature Selection Process.* The preprocessed text data was transformed into numerical feature vectors using TF-IDF. Simultaneously, *n*-grams were extracted and included as additional features. The final feature vectors for each document were constructed by combining TF-IDF scores with *n*-gram frequencies. This comprehensive feature set ensures that both individual term importance and contextual patterns are considered in the classification models.

The integration of TF-IDF and n-grams into the feature selection process enhances the model's ability to accurately identify and classify destructive speech in Kazakh. By capturing both term significance and contextual patterns, these features contribute to the overall effectiveness of the detection system.

While our study emphasizes the importance of language-specific modeling for low-resource languages like Kazakh, we chose to use mBERT over KazBERT due to several critical factors relating to data coverage, generalization capability, and implementation stability. KazBERT, although developed specifically for Kazakh, was trained on a relatively small and localized dataset comprising only 5,904 Kazakh Wikipedia texts, along with limited English and Russian corpora (*Amandyk, 2023*). This narrow scope reduces the robustness and generalizability of the resulting language representations, especially for tasks that require diverse contextual understanding beyond formal Wikipedia-style language. Moreover, the model's trilingual design (Kazakh, Russian, English) introduces a potential risk of overfitting to the styles of those domains, making it less suitable for broad applications.

In contrast, mBERT was pretrained on the Wikipedias of 104 languages and includes over 200,000 Kazakh Wikipedia entries at the time of training. While it may not specialize in Kazakh, its broad multilingual context supports better cross-lingual transfer and

**Table 3 Used ML algorithms.**

| Machine learning technique | Description |
|---|---|
| Logistic regression | Used for binary and multiclass classification tasks, logistic regression is employed to model the probability of each type of destructive speech based on TF-IDF and n-gram features. It provides a baseline =for comparing the performance of more complex models. |
| Support vector machine (SVM) | SVM creates optimal hyperplanes for separating different classes of text. By using TF-IDF and *n*-gram features, SVM effectively classifies text into categories such as religious extremism, bullying, national extremism, and racism. |
| Naive Bayes | This probabilistic classifier applies Bayes' theorem with strong independence assumptions between features. It is useful for handling high-dimensional feature spaces and provides a baseline for text classification. |
| Random forest | An ensemble method that aggregates the results of multiple decision trees. It improves classification accuracy and robustness by averaging predictions and reducing overfitting. |
| Gradient boosting | This technique builds models sequentially, with each new model correcting errors made by previous ones. It enhances performance by focusing on difficult-to-classify instances and integrating TF-IDF and n-gram features. |
| AdaBoost | Adaptive Boosting combines multiple weak classifiers to create a strong classifier. It improves classification performance by focusing on incorrectly classified instances and adjusting weights accordingly. |
| LSTM1 | This LSTM network is designed to capture long-term dependencies and contextual information in the text, addressing the challenges of sequential data. |
| LSTM2 | An enhanced LSTM variant with additional layers or units to capture more complex patterns and improve performance in detecting destructive speech. |
| Gated Recurrent Unit (GRU) | GRU is employed as an alternative to LSTM, providing similar benefits in capturing sequential dependencies with a simplified architecture that can be more computationally efficient. |
| Convolutional neural network + LSTM (CNN+LSTM) | This hybrid model combines CNNs, which capture local patterns and hierarchical features, with LSTM networks, which model sequential dependencies. This combination enhances the model's ability to understand both local and global contexts in the text. |
| Ensemble method | To leverage the strengths of individual models and improve overall performance, predictions from various machine learning and deep learning models are combined. This approach integrates results from logistic regression, SVM, Naive Bayes, random forest, gradient boosting, AdaBoost, and the deep learning methods, enhancing the model's robustness and accuracy in detecting destructive speech. |

generalization. Studies such as *Wu & Dredze (2020)* suggest that multilingual models benefit low-resource languages through shared representations across typologically similar languages. Furthermore, mBERT offers a stable, well-integrated pipeline within the Hugging Face ecosystem, whereas KazBERT's documentation and support are still limited. Ultimately, the choice of mBERT was driven by practical considerations regarding model maturity, infrastructure support, and the broader linguistic diversity captured during pretraining, which compensates, to an extent, for its Kazakh-specific limitations. Future work could include a controlled empirical comparison between mBERT and KazBERT on downstream Kazakh NLP tasks.

In this study, we adopt two complementary approaches to destructive speech detection: (1) classical machine learning using feature-based methods and (2) a deep learning architecture combining BERT embeddings with an LSTM layer. This dual strategy enables a comprehensive evaluation of both traditional and modern techniques for Kazakh-language content classification (Table 3).

The machine learning models selected for this study—Naive Bayes, logistic regression, random forest and others—are widely recognized as strong baseline algorithms in text classification tasks, particularly in the domains of hate speech detection, cyberbullying identification, and toxic content moderation (*Schmidt & Wiegand, 2017*; *Fortuna & Nunes, 2018*; *Badjatiya et al., 2017*). Naive Bayes is known for its effectiveness in handling high-dimensional sparse feature spaces generated by TF-IDF representations. Logistic regression has consistently demonstrated competitive performance in binary and multi-class text classification scenarios, offering a balance between simplicity and accuracy. random forest provides robustness through ensemble learning, capturing complex feature interactions without overfitting easily.

For the deep learning approach, the combination of BERT embeddings with an LSTM layer was motivated by the need to model both contextual semantics and sequential dependencies, particularly important for the morphologically rich Kazakh language. BERT-based models have achieved state-of-the-art results in various NLP tasks (*Devlin et al., 2019*), while LSTM networks are effective in capturing long-range dependencies in sequential data (*Hochreiter & Schmidhuber, 1997*).

The integration of an LSTM layer following BERT embeddings in our model is motivated by prior research demonstrating the effectiveness of hybrid transformer-RNN architectures for capturing sequential dependencies in morphologically complex languages. While BERT models capture contextualized representations through self-attention, they are limited in modeling fine-grained morphological variations and long-distance sequential dependencies, which are critical in agglutinative languages such as Kazakh.

The article of *Gethsia, Juliet & Anitha (2024)* presents a hybrid model that combines BERT for contextual embeddings with Bidirectional LSTM (BiLSTM) for sequential modeling to enhance emotion recognition in text. This approach captures both the deep contextual meaning of words and their order in a sentence. The model outperforms standalone BERT and BiLSTM architectures on benchmark datasets. It shows strong potential for applications like sentiment analysis and emotional tone detection in social media and customer feedback.

The study explores the use of BERT-based embeddings combined with hybrid neural network architectures for Indonesian sentiment analysis. It highlights that BERT significantly improves performance by capturing contextual nuances in the language. Among the tested models, the BERT-based LSTM-CNN achieved the highest accuracy. The research demonstrates that combining BERT with hybrid deep learning models is highly effective for sentiment classification in low-resource languages like Indonesian (*Murfi et al., 2024*).

In the article "Using BERT and LSTM to Do Text Classification", *Lu (2022)* addresses the challenge of accurately interpreting multiword expressions (MWEs) in natural language processing. Recognizing that pre-trained models like BERT may struggle with the non-compositional meanings of MWEs, Lu proposes enhancing BERT with an additional

LSTM layer to better capture sequential dependencies. This hybrid BERT-LSTM model demonstrated improved performance in text classification tasks, outperforming both standalone BERT and other hybrid configurations such as BERT with TextCNN. The study underscores the effectiveness of integrating LSTM with BERT to enhance the understanding of complex linguistic structures in text classification?.

In our context, the LSTM layer serves to further refine the contextual sequence representations generated by BERT, particularly for Kazakh's rich case system, verb morphology, and word order variability.

To evaluate the impact of different textual representations on model performance, we systematically experimented with multiple n-gram configurations across both machine learning and deep learning models. Specifically, we used three n-gram ranges: ($i$) $N = 1, 2$, which includes both unigrams and bigrams to capture single-word tokens and adjacent word pairs; ($ii$) $N = 2, 2$, which uses only bigrams to emphasize local word pair dependencies; and ($iii$) $N = 2, 3$, which combines bigrams and trigrams to represent richer multi-word contextual patterns. Each configuration was tested under two conditions: with stemming and without stemming. Stemming reduces words to their morphological roots, helping to normalize inflectional variations common in Kazakh and thereby reducing feature sparsity. In contrast, the no-stemming condition preserves the original surface forms of words, allowing the models to capture exact lexical usage patterns. These combined configurations were designed to assess the sensitivity of various classifiers to different levels of lexical and contextual abstraction in Kazakh texts. As shown in Tables 4–9, the $N = 1, 2$ with stemming setting consistently yielded the best overall results across multiple evaluation metrics, indicating that the combination of lexical normalization and local context features is particularly effective for destructive speech classification in morphologically rich languages such as Kazakh.

## Deep learning pipeline: BERT+LSTM hybrid model

For the deep learning pipeline, we designed a hybrid architecture combining pre-trained BERT embeddings with a sequential modeling component based on an LSTM layer. The motivation for this design stems from the observation that while BERT embeddings capture rich contextual information at the token level, they may not fully model the sequential structure of texts, especially for languages like Kazakh, which exhibit complex morphological patterns.

In the proposed architecture, the input texts are first transformed into contextual embeddings using a pre-trained BERT model. These embeddings are then passed through a unidirectional LSTM layer to model sequential dependencies between words. The output of the LSTM layer is subsequently fed into dense layers with ReLU activations, followed by a softmax output layer for multi-class classification. These layers were chosen to effectively capture both the contextual and sequential characteristics of the Kazakh language. BERT embeddings provide deep contextual understanding, LSTM models sequential dependencies that transformers alone may overlook, and the dense layer maps the learned representations to the target classes for accurate classification. The model was trained

**Table 4 Performance comparison of machine learning and deep learning models for destructive speech classification in Kazakh ($N = 1, 2$ with stemming).**

| $N = 1, 2$ with stemming | Accuracy | F1-score | AUC-ROC |
|---|---|---|---|
| **Machine learning methods** | | | |
| Logistic regression | 0.8685 | 0.8665 | 0.9760 |
| SVM | 0.8701 | 0.8681 | 0.9751 |
| Naive Bayes | 0.7759 | 0.7556 | 0.9718 |
| Random forest | 0.7629 | 0.7664 | 0.9506 |
| Gradient boosting | 0.7378 | 0.7370 | 0.9370 |
| Ensemble | 0.8649 | 0.8631 | 0.9774 |
| Adaboost | 0.8139 | 0.8169 | 0.9635 |
| **Deep learning methods** | | | |
| LSTM1 | 0.8708 | 0.8721 | 0.9769 |
| LSTM2 | 0.8665 | 0.8674 | 0.9777 |
| GRU | 0.8543 | 0.8565 | 0.9740 |
| CNN+LSTM | 0.8296 | 0.8312 | 0.9599 |

using the Adam optimizer with a learning rate of $1e-4$, and early stopping was applied based on validation loss to prevent overfitting.

### Embedding layer

The embedding layer is an important part of the proposed model architecture, which is used to transform the text into numeric vectors, which are then used further. We used pre-trained embeddings specifically designed for the Kazakh language to preserve its vocabulary and grammatical structure. Using multiple embeddings allows the model to capture the relationships between words, which are necessary for proper understanding of the context.

In this study, we made sure that the text length did not exceed one hundred and twenty-eight characters. This choice was made in order to reduce computational costs, as well as to ensure the uniformity of the input data. In case the text size is smaller than its maximum size, it is padded with special filler markers, and in case its maximum size is exceeded, it is reduced to a limited maximum size. In this way, each piece of text is compressed to a certain uniform length, which facilitates subsequent processes.

The input text first passes through the embedding layer, which transforms the original text into numeric vectors that the machine learning model can interpret. Embedding is a way of representing words as dense vectors of real numbers that captures the semantic relationships between words. The size of the embedding vector depends on the length of each input text. After tokenization (breaking the text into words or subwords), each token is matched with the corresponding embedding vector.

**Table 5 Performance comparison of machine learning and deep learning models for destructive speech classification in Kazakh ($N = 1, 2$ without stemming).**

| $N = 1, 2$ no stemming | Accuracy | F1-score | AUC-ROC |
| --- | --- | --- | --- |
| **Machine learning methods** | | | |
| Logistic regression | 0.8296 | 0.8276 | 0.9668 |
| SVM | 0.8579 | 0.8561 | 0.9692 |
| Naive Bayes | 0.7523 | 0.7372 | 0.9663 |
| Random forest | 0.7394 | 0.7410 | 0.9312 |
| Gradient boosting | 0.6974 | 0.6936 | 0.9194 |
| Ensemble | 0.8273 | 0.8255 | 0.9676 |
| Adaboost | 0.7872 | 0.7891 | 0.9513 |
| **Deep learning methods** | | | |
| LSTM1 | 0.8473 | 0.8477 | 0.9742 |
| LSTM2 | 0.8477 | 0.8474 | 0.9725 |
| GRU | 0.8273 | 0.8323 | 0.9657 |
| CNN+LSTM | 0.8198 | 0.8201 | 0.9594 |

**Table 6 Performance comparison of machine learning and deep learning models for destructive speech classification in Kazakh ($N = 2, 2$ with stemming).**

| $N = 2, 2$ with stemming | Accuracy | F1-score | AUC-ROC |
| --- | --- | --- | --- |
| **Machine learning methods** | | | |
| Logistic regression | 0.6534 | 0.6450 | 0.9071 |
| SVM | 0.6601 | 0.6505 | 0.9062 |
| Naive Bayes | 0.6212 | 0.6009 | 0.9067 |
| Random forest | 0.5714 | 0.5827 | 0.9067 |
| Gradient boosting | 0.5192 | 0.5006 | 0.8041 |
| Ensemble | 0.6946 | 0.6925 | 0.9005 |
| Adaboost | 0.5486 | 0.5405 | 0.8728 |
| **Deep learning methods** | | | |
| LSTM1 | 0.6558 | 0.6593 | 0.8963 |
| LSTM2 | 0.6381 | 0.6387 | 0.8943 |
| GRU | 0.5883 | 0.5958 | 0.8583 |
| CNN+LSTM | 0.5400 | 0.5449 | 0.8158 |

**Table 7 Performance comparison of machine learning and deep learning models for destructive speech classification in Kazakh ($N = 2, 2$ without stemming).**

| $N = 2, 2$ no stemming | Accuracy | F1-score | AUC-ROC |
| --- | --- | --- | --- |
| **Machine learning methods** | | | |
| Logistic regression | 0.6538 | 0.6428 | 0.9077 |
| SVM | 0.6577 | 0.6465 | 0.9075 |
| Naive Bayes | 0.6110 | 0.5933 | 0.9063 |

*(Continued)*

| Table 7 (continued) | | | |
| --- | --- | --- | --- |
| N = 2, 2 no stemming | Accuracy | F1-score | AUC-ROC |
| Random forest | 0.5357 | 0.5401 | 0.8271 |
| Gradient boosting | 0.5180 | 0.5101 | 0.8062 |
| Ensemble | 0.6942 | 0.6908 | 0.9063 |
| Adaboost | 0.5164 | 0.5037 | 0.8645 |
| **Deep learning methods** | | | |
| LSTM1 | 0.6251 | 0.6346 | 0.8724 |
| LSTM2 | 0.5624 | 0.5364 | 0.8542 |
| GRU | 0.5569 | 0.5722 | 0.8307 |
| CNN+LSTM | 0.5364 | 0.5551 | 0.8067 |

**Table 8 Performance comparison of machine learning and deep learning models for destructive speech classification in Kazakh (N = 2, 3 with stemming).**

| N = 2, 3 with stemming | Accuracy | F1-score | AUC-ROC |
| --- | --- | --- | --- |
| **Machine learning methods** | | | |
| Logistic regression | 0.6491 | 0.6384 | 0.9054 |
| SVM | 0.6601 | 0.6497 | 0.9049 |
| Naive Bayes | 0.6098 | 0.5855 | 0.9050 |
| Random forest | 0.5518 | 0.5623 | 0.8311 |
| Gradient boosting | 0.4960 | 0.4800 | 0.8024 |
| Ensemble | 0.6934 | 0.6903 | 0.8939 |
| Adaboost | 0.5404 | 0.5309 | 0.8717 |
| **Deep learning methods** | | | |
| LSTM1 | 0.6397 | 0.6414 | 0.8889 |
| LSTM2 | 0.6593 | 0.6667 | 0.8876 |
| GRU | 0.6401 | 0.6527 | 0.8734 |
| CNN+LSTM | 0.5596 | 0.5634 | 0.8203 |

**Table 9 Performance comparison of machine learning and deep learning models for destructive speech classification in Kazakh (N = 2, 3 without stemming).**

| N = 2, 3 no stemming | Accuracy | F1-score | AUC-ROC |
| --- | --- | --- | --- |
| **Machine learning methods** | | | |
| Logistic regression | 0.6471 | 0.6335 | 0.9056 |
| SVM | 0.6526 | 0.6411 | 0.9064 |
| Naive Bayes | 0.6036 | 0.5836 | 0.9051 |
| Random forest | 0.5392 | 0.5422 | 0.8161 |
| Gradient boosting | 0.5027 | 0.4848 | 0.7888 |
| Ensemble | 0.6911 | 0.6871 | 0.9038 |
| Adaboost | 0.5078 | 0.4915 | 0.8626 |
| **Deep learning methods** | | | |
| LSTM1 | 0.5963 | 0.5990 | 0.8528 |

| $N = 2, 3$ no stemming | Accuracy | F1-score | AUC-ROC |
|---|---|---|---|
| LSTM2 | 0.5800 | 0.5890 | 0.8538 |
| GRU | 0.5361 | 0.5532 | 0.8092 |
| CNN+LSTM | 0.5027 | 0.5037 | 0.7911 |

The text embedding process can be described as follows. Let $T = \{w_1, w_2, \ldots, w_n\}$—be a sequence of tokens representing the text, where $n \neq 128$. Each token $v_i$ is transformed into a vector representation $v_i$ using the embedding function $E$:

$$v_i = E(w_i), \quad v_i = R^d \tag{5}$$

where $d$ is the dimension of the embedding space. As a result, the text $T$ is transformed into a sequence of vectors $V = v_1, v_2, \ldots, v_n$.

These vectors convey information about the words in relation to the rest of the sentence, and this is crucial for understanding the Kazakh language due to its complexity. It is also possible to address the analysis of the text, rather than its individual elements, thanks to the use of pre-trained embeddings.

It should be noted that since the input data is fixed in length, this allows for the use of parallel computing, which significantly increases the speed of model training. This also increases the stability of the network, since all input data has the same shape.

It should be noted that since the input data is fixed in length, this allows parallel computing to be used, which significantly increases the speed of model training. It also increases the stability of the network, since all input data is of the same shape.

### BERT layer

BERT, or Bidirectional Encoder Representation Transformer (BERT), is a recent and remarkable approach to solving NLP problems due to its ability to take into account the context of words in text in both directions. This is important for a language with rich morphology and flexible word order, such as Kazakh, where many forms are context-dependent. The central innovation of BERT is the transformer architecture, which first processes text using attention mechanisms at a large number of levels and can also capture complex inter-word relationships.

BERT is based on the transformer architecture, which consists of encoder layers. Each encoder layer includes two sublayers, *i.e.*, a self-supervision mechanism and a feedforward neural network, which are connected more efficiently using residual connections that surround each of them as follows; definition of the self-supervision mechanism:

$$Attention(Q, K, V) = softmax\left(\frac{QK^T}{\sqrt{d_k}}\right)V. \tag{6}$$

But the fundamental difference is that here, by adding the $d_k$ dimension as part of the softmax denominator, we not only make it more stable (as discussed earlier), but also invalidate the pseudo-dimension for $Q$ and $K$ before the attention values are multiplied together.

The matrices $Q$ (query), $K$ (key), and $V$ (value) in our case are derived from the input embeddings, and $d_k$ is also a key vector dimension. This attention mechanism allows the model to highlight different words with respect to the entire sentence, which allows us to obtain a rich, contextualized vector representation of each word.

The attention mechanism is particularly useful in Kazakh, as the word order often changes almost effortlessly in Kazakh, and it allows the model to pay special attention to some parts of the sentence and not others, regardless of whether the keywords are at the beginning or not. In Kazakh, for example, the subject-object-verb order is not strict, and changes in meaning occur due to the morphology of the word, such as "enough". BERT's self-supervision mechanism copes well with this ambiguity, as it can dynamically learn to focus on what is most relevant based on the context.

Once the text is embedded, it is passed through a pre-trained BERT model. BERT is designed to capture both left and right context simultaneously, making it particularly effective for languages like Kazakh that have flexible word order.

The BERT model generates contextualized embeddings for the input text with a fixed size of $N \times 768$. This means that no matter how long the input text is, BERT outputs embeddings in which each token is represented by a 768-dimensional vector. The pre-trained BERT model is used to reduce training time and improve the understanding of the nuances of the Kazakh text.

After the self-attention operation, each encoder layer applies a feedforward neural network, which processes the output of the attention mechanism. This network helps to refine the learned representations and introduce non-linearity, enhancing the model's ability to capture complex patterns in the text.

A key innovation of BERT is its bidirectional nature. Unlike traditional NLP models such as Word2Vec or GloVe, which generate static word embeddings based on the context of surrounding words either before or after the target word, BERT considers both directions simultaneously. This means that BERT looks at the full context of a word, using both the left and right sides of the sentence to generate a contextualized embedding.

In languages like Kazakh, where the meaning of a word can change drastically based on its position and accompanying morphological markers, this bidirectional approach is crucial. For example, a word in Kazakh may have different meanings depending on its suffixes or neighboring words. By analyzing the sentence from both directions, BERT can better understand these nuanced dependencies and generate a more accurate representation of each word in its specific context. This bidirectional contextualization is achieved through BERT's pre-training process, which uses two objectives:

- Masked Language Modeling (MLM): BERT randomly masks some of the words in the input and trains the model to predict the masked words using the surrounding context from both directions.
- Next Sentence Prediction (NSP): BERT is also trained to predict whether one sentence logically follows another, helping it understand relationships between sentences.

The MLM task is especially useful in Kazakh, where word order and suffixes can vary, as it forces the model to rely on the entire sentence context to make predictions. The NSP task further enhances BERT's ability to understand longer-range dependencies between sentences, which is valuable for detecting destructive speech patterns that may span multiple sentences.

Traditional NLP models, such as Word2Vec or GloVe, generate static word embeddings where each word is represented by a fixed vector, regardless of its context in the sentence. These models do not consider how the meaning of a word changes based on its neighbors or position, which can be a significant limitation, particularly for languages like Kazakh with flexible word order and rich morphology.

In contrast, BERT generates dynamic embeddings, where each word's representation is influenced by the specific context in which it appears. This allows BERT to capture subtle changes in meaning that static models cannot. For example, the word "келді" in Kazakh, which means "came," may be interpreted differently depending on the subject and object, and BERT's bidirectional architecture can fully capture these dependencies by looking at both the preceding and following words.

Another limitation of traditional models is their inability to handle long-range dependencies effectively. Word2Vec and GloVe rely on a fixed window size for context, meaning they struggle with sentences where critical relationships exist between distant words. BERT overcomes this by using the transformer's self-attention mechanism, which allows the model to weigh all words in the sentence when generating embeddings, regardless of their distance from each other.

### LSTM layers

The contextualized embeddings from BERT are then fed into a long short-term memory (LSTM) network (Fig. 3). LSTM is a type of recurrent neural network (RNN) that is well-suited for sequence data, such as text, as it can capture long-term dependencies within the sequence.

LSTM helps to further refine the features learned by BERT by modeling the sequential nature of the text. In this architecture:

- LSTM layers $N \times 128 \times 256$: the input to the LSTM is the sequence of embeddings from BERT, and the LSTM transforms this into an output of size $N \times 128 \times 256$. Here, 128 refers to the hidden layer size, and 256 refers to the output dimension of the LSTM.
- The LSTM layer processes the data in time steps, sequentially processing the embeddings from BERT and allowing the model to capture time-dependencies (or sequential information) that may exist in the text.

The LSTM layer further enhances the model's ability to capture dependencies in text sequences, which can help in tasks where understanding the order of words or sentences is crucial, such as in distinguishing between neutral and extremist content.

An LSTM layer was used after the BERT embeddings to enhance the model's ability to capture sequential dependencies within Kazakh texts. Although transformer models like BERT are highly effective in modeling token-level contextual relationships, they may not

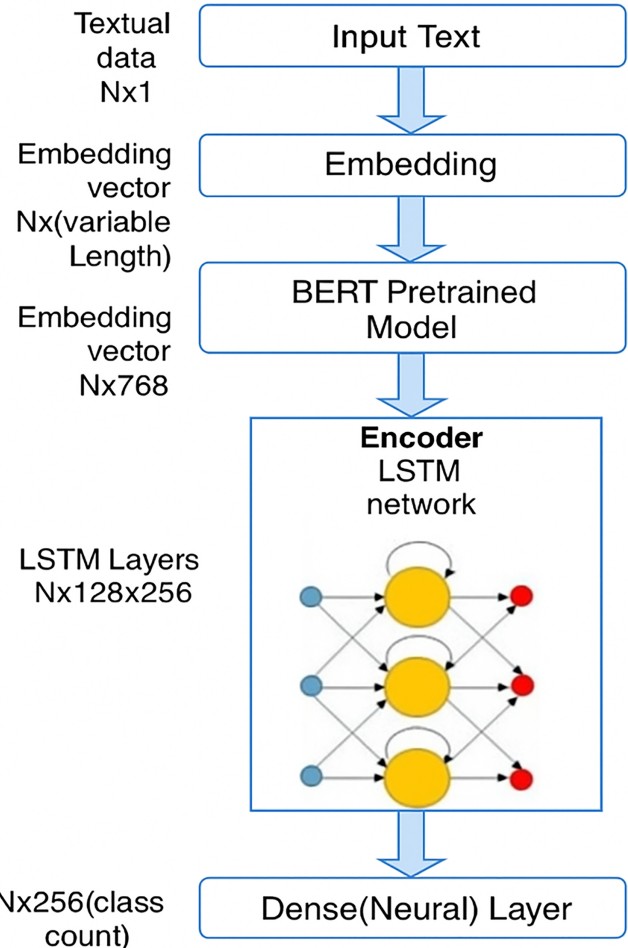

Textual
data
Nx1

Embedding
vector
Nx(variable
Length)

Embedding
vector
Nx768

LSTM Layers
Nx128x256

Nx256(class
count)

**Figure 3 Architecture diagram of the proposed model.**

fully preserve longer sequential patterns, especially in languages with rich morphology and flexible syntax like Kazakh. LSTM networks are well-suited for learning long-term dependencies and sequential structures, making them an appropriate choice for improving classification accuracy in this setting. Alternative models such as GRU or transformer encoders were considered; however, LSTM was selected due to its proven effectiveness in capturing complex sequence patterns with relatively lower computational overhead.

### Dense layer

After passing through the LSTM layer, the output is forwarded to a dense (neural) layer. The role of this layer is to perform the final classification task.

- $N \times 256$ (class count): the dense layer has an output dimension of 256, which corresponds to the number of possible output classes. In the case of Kazakh text classification, the class count could represent categories such as neutral, propaganda, recruitment, radicalization, *etc.*

- This layer performs the classification using the features extracted by both the BERT and LSTM layers.

The architecture presented in Fig. 3 illustrates the operational workflow of our BERT+LSTM model. In this model, contextual embeddings generated by the pretrained BERT layer are passed into the LSTM network, which captures sequential dependencies within the Kazakh text. The output of the LSTM is then forwarded to a final classification (dense) layer. This architecture was specifically applied in the experiments for destructive content classification shown in Fig. 4. In contrast, the results presented in Tables 4–9, described in "Experimental Results", were obtained using traditional machine learning (ML) and standalone deep learning (DL) models, and therefore are not based on the BERT +LSTM architecture shown in Fig. 3.

### Evaluation metrics

To assess the performance of our model for detecting destructive speech in Kazakh, we utilized several key evaluation metrics. These metrics provide a comprehensive view of the model's effectiveness in classification tasks. The primary metrics used are:

accuracy measures the proportion of correctly classified instances out of the total number of instances. It provides an overall sense of how well the model is performing.

$$Accuracy = \frac{TP + TN}{TP + TN + FP + FN}. \tag{7}$$

- TP (True Positives)—the number of correctly predicted positive instances;
- TN (True Negatives)—number of correctly predicted negative instances;
- FP (False Positives)—the number of falsely predicted positive instances;
- FN (False Negatives)—number of wrongly predicted negative instances.

Precision evaluates the proportion of true positive predictions relative to the total number of positive predictions made by the model. It indicates how many of the identified positive instances are actually positive.

$$Precision = \frac{TP}{TP + FP}. \tag{8}$$

Recall (or Sensitivity) measures the proportion of true positive predictions relative to the total number of actual positive instances. It indicates how well the model identifies all relevant instances.

$$Recall = \frac{TP}{TP + FN} \tag{9}$$

The *F*1 Score is the harmonic mean of precision and recall, providing a single metric that balances both aspects. It is particularly useful when dealing with imbalanced datasets where both false positives and false negatives are critical.

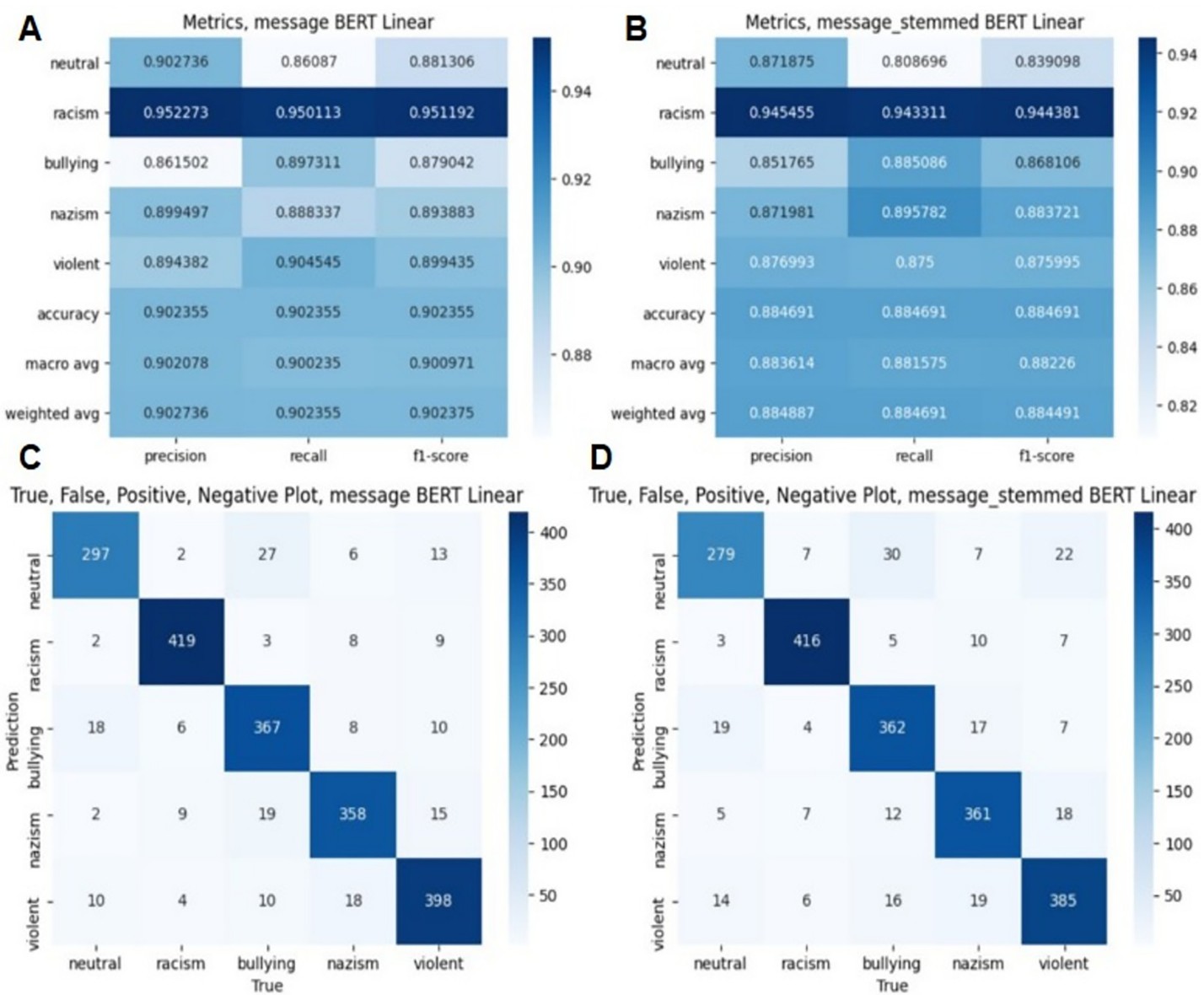

**Figure 4 Evaluation metrics in BERT algorithm.** (A) Classification report on without stemming experiment. (B) Classification report on with stemming experiment. (C) True, False means on without stemming experiment. (D) True, False means on with stemming experiment.

$$F1\text{-}score = 2 \times \frac{Precision \times Recall}{Recall + Precision}. \tag{10}$$

Area under the receiver operating characteristic curve (AUC-ROC) measures the ability of the model to distinguish between classes. The ROC curve plots the true positive rate against the false positive rate at various thresholds, and AUC represents the area under this curve. A higher AUC indicates better model performance. An AUC value closer to 1 signifies excellent performance, while a value closer to 0.5 indicates performance equivalent to random guessing.

The confusion matrix provides a detailed breakdown of the model's predictions, showing the counts of true positives, true negatives, false positives, and false negatives. It helps in understanding the types of errors the model is making.

These metrics collectively offer insights into the model's performance in detecting various types of destructive speech, including its accuracy, precision, recall, and ability to handle imbalanced classes. Evaluating the model using these metrics ensures a comprehensive assessment of its effectiveness and helps in fine-tuning and optimizing its performance.

# EXPERIMENTAL CONFIGURATIONS

In our experiments, we used the BERT model, specifically the bert-base-multilingual-uncased variant, which is pretrained on multiple languages and capable of handling the diverse nature of the input data. For optimal performance, we fixed several critical hyperparameters as follows:

- Max sequence length: we set the maximum sequence length to 128 tokens. This allows for capturing sufficient contextual information while maintaining computational efficiency.
- Batch size: we trained the model with a batch size of 128, balancing memory consumption and gradient stability during training.
- Number of epochs: the model was trained for 10 epochs, providing enough iterations to converge while avoiding overfitting.
- Learning rate: we employed a learning rate of $2e-5$, a typical value for fine-tuning transformer-based models, allowing for stable convergence during training.
- Dropout rate: a dropout rate of 0.1 was applied to prevent overfitting, ensuring that the model does not rely too heavily on specific neurons during training.
- Scheduler: to ensure smooth training, we used a linear learning rate scheduler with warm-up steps, which gradually decreases the learning rate as training progresses. This helps to prevent abrupt updates in the model's weights early on, enhancing the convergence process.
- Device: the model was trained and evaluated on a GPU (cuda) if available; otherwise, it fell back to using a CPU for processing.
- Random seed: to ensure reproducibility across different runs, we fixed a random seed of 42 for the entire pipeline, including data loading, model initialization, and optimization processes.

These hyperparameter choices were guided by empirical findings from prior work with BERT-based models and were refined to suit the characteristics of our dataset. The combination of these settings allowed for the effective learning and generalization of the model across the diverse input data.

## EXPERIMENTAL RESULTS

In this section, we present the results of our evaluation for the model designed to detect destructive speech in Kazakh. The performance of various machine learning and deep learning algorithms is assessed using the metrics previously defined.

Figure 4 and Table 4 demonstrate classification results in BERT algorithm. Figure 4 demonstrates the classification results of the BERT-based model, while Table 4 presents comparative evaluation results of various machine learning and deep learning methods, including BERT+LSTM.

The results show that machine and deep learning methods show different performance in data classification when using $N = 1, 2$ with stemming. Metrics such as accuracy, F1-score and AUC-ROC vary depending on the algorithm used. The SVM and logistic regression models had the highest results for all indicators. This result indicates that the methods have a good balance between precision and recall. Deep learning systems have also performed well. For example, the LSTM model performed best among all methods. Naive Bayes and random forest methods showed relatively low results, which may indicate their low ability to effectively process data in this task. Ensemble method and Adaboost also performed well, but they performed slightly lower than individual methods such as SVM and LSTM. In general, based on these data, it can be concluded that the best results are achieved by using SVM and LSTM models for the classification task (Table 4).

The results show the effectiveness of different machine and deep learning methods for data classification using $N = 1.2$ no stemming. Among the machine learning methods, SVM and logistic regression have the best accuracy and F1-scores, indicating their ability to perform the task well. Naive Bayes also shows significantly higher AUC-ROC values despite lower precision and F1-measure values. Deep learning methods also show competitive results. LSTMs show high performance in all indicators, especially AUC-ROC, which indicates their good recognition ability in data. Both GRU and CNN+LSTM performed better than LSTM methods, although they were relatively lower in terms of accuracy and F1-measure (Table 5).

According to the results of using $N = 2.2$ with stemming, accuracy indicators and F1 measurements for all models in the machine learning method decreased significantly. SVM and logistic regression performed best among machine learning methods. Naive Bayes and random forest also have similar AUC-ROC values, but their accuracy and F1-measure are significantly lower, indicating difficulties in correctly classifying the data. Gradient boosting and Adaboost showed the least satisfactory results.

LSTM models in deep learning methods still maintain relatively high values, but their accuracy and F1-measure are at 0.65, which indicates their reduced efficiency in this configuration. GRU and CNN + LSTM performed even worse, especially in AUC-ROC, indicating their difficulty in distinguishing between classes (Table 6).

According to the results of using $N = 2.2$ no stemming, machine learning methods are showing normal results. SVM and logistic regression remain leaders in terms of accuracy and F1-measure. Naive Bayes shows slightly worse results, especially in terms of accuracy and F1-score, but maintains a high AUC-ROC, indicating a good ability of the model to

recognize classes. Random forest and Gradient boosting significantly reduce the accuracy and F1 measures, indicating their low efficiency in this case.

Deep learning techniques have shown performance degradation. LSTM models, although competitive, still showed a significant decrease in accuracy and F1-score, especially LSTM2, which performed the worst among LSTMs. GRU and CNN+LSTM were also worse. Overall, the results show a significant reduction in accuracy and F1-score compared to previous experiments. Deep learning methods show the largest drop in performance, while machine learning and ensemble methods remain robust despite limited performance (Table 7).

According to the classification models using $N = 2.3$ with stemming, despite different approaches, the overall performance of all models was average.

Machine learning algorithms such as SVM and logistic regression show an accuracy of around 65%. Combining models leads to some performance improvement, although no significant performance increase is observed.

Deep learning methods like LSTM and GRU also show similar results with around 64–66% accuracy. CNN+LSTM performed slightly worse than other deep learning models (Table 8).

Results using $N = 2.3$ with stemming for both machine and deep learning models show that none of the models are significantly different from the others, while overall results remain average.

Machine learning methods such as SVM and logistic regression show the best results with an accuracy of about 65% and an AUC-ROC of about 0.90. Combining models improves performance slightly. The weakest results are shown by Gradient boosting and Adaboost models, with accuracies slightly higher than 50% and lower AUC-ROC.

Deep learning methods such as LSTM and GRU show similar results with accuracies in the range of 64–66%. The CNN+LSTM model shows poor performance compared to other deep learning methods (Table 9).

In addition to the traditional and deep learning models discussed earlier, Table 10 presents the evaluation results of advanced transformer-based models and graph neural networks. BERT and Cross-lingual Language Model with Masked Language Modeling (XLM-MLM) were selected due to their high performance in multilingual text classification tasks. These models were tested with and without stemming to observe the impact of morphological preprocessing on classification accuracy.

The results of the deep learning methods show a significant improvement in accuracy compared to the above models.

BERT and XLM-MLM show high accuracy rates of 90% and 89%, respectively, indicating their excellent ability to classify data. When using stemming, the accuracies of the BERT and XLM-MLM models decrease to 88.47% and 88.96%, respectively, but they are still higher than the other approaches.

In conclusion, various models of deep learning techniques with machine learning techniques have been developed to detect destructive messages. A number of experiments were conducted on the proposed models. As a result of the conducted experiments, a model for detecting destructive messages on web resources was proposed. In comparison,

**Table 10 Accuracy indicators of various deep learning architectures (BERT, XLM-MLM) on Kazakh-language destructive content classification.**

**Deep learning methods**

| No stemming | Accuracy | F1-score | AUC-ROC |
|---|---|---|---|
| BERT | 0.9024 | 0.9024 | 0.9816 |
| XLM-MLM | 0.8911 | 0.8909 | 0.9791 |
| **with stemming** | **Accuracy** | **F1-score** | **AUC-ROC** |
| BERT | 0.8847 | 0.8845 | 0.9817 |
| XLM-MLM | 0.8896 | 0.8899 | 0.9807 |

the highest accuracy in machine learning models $N = 1.2$ with stemming was logistic regression (0.8685), SVM (0.8701), Ensemble (0.8638), and the highest accuracy in deep learning models without stemming (no stemming) BERT (0.9024) showed a high index.

## CONCLUSION AND FUTURE WORK

In this study, we addressed the critical issue of detecting destructive speech in the Kazakh language, focusing on four key types: religious extremism, bullying, national extremism, and racism. By leveraging a combination of machine learning and deep learning techniques, we aimed to create a robust model capable of accurately identifying and classifying harmful content in Kazakh text data.

Our approach involved the use of a diverse set of algorithms, including logistic regression, support vector machines (SVM), Naive Bayes, random forest, gradient boosting, and AdaBoost, which provided valuable benchmarks and insights into the classification performance. These traditional machine learning methods demonstrated commendable accuracy and effectiveness in detecting destructive speech, each contributing uniquely to the overall performance.

We also incorporated advanced deep learning methods, such as long short-term memory (LSTM) networks, Gated Recurrent Units (GRU), and convolutional neural networks combined with LSTM (CNN+LSTM). These models excelled in capturing complex patterns and contextual information in the text, proving particularly useful in understanding the sequential and contextual nature of the data.

The study employed various n-gram configurations and stemming techniques to enhance feature extraction. The use of unigrams, bigrams, and trigrams, both with and without stemming, allowed the model to effectively capture linguistic patterns and contextual nuances. This approach improved the model's ability to detect and classify harmful speech by providing a comprehensive representation of the text data.

The ensemble method, which combined the predictions of multiple models, achieved superior performance, demonstrating the effectiveness of integrating different techniques to address the challenges of destructive speech detection. By leveraging the strengths of individual models, the ensemble approach provided a more accurate and robust classification solution.

Our findings highlight the importance of using a multifaceted approach in the detection of destructive speech. The combination of machine learning and deep learning techniques, along with diverse feature extraction methods, has proven to be effective in improving classification accuracy and handling the complexities of Kazakh text data.

Looking ahead, there are several avenues for future research. Enhancing the model's performance through additional feature engineering, exploring new deep learning architectures, and addressing any limitations identified in this study could further refine the model's accuracy and applicability. Moreover, expanding the dataset and incorporating more diverse linguistic resources could improve the model's robustness and generalization capabilities.

In summary, this study contributes to the field of harmful content detection by providing a comprehensive evaluation of various machine learning and deep learning methods for the Kazakh language. The insights gained from this research offer valuable guidance for developing more effective systems for identifying and mitigating destructive speech in digital communications.

### Funding
This work was supported by the Ministry of Science and Higher Education of the Republic of Kazakhstan. Grant No. IRN AP19676342. The funders had no role in study design, data collection and analysis, decision to publish, or preparation of the manuscript.

### Grant Disclosures
The following grant information was disclosed by the authors:
Ministry of Science and Higher Education of the Republic of Kazakhstan: IRN AP19676342.

### Competing Interests
The authors declare that they have no competing interests.

### Author Contributions
- Milana Bolatbek conceived and designed the experiments, analyzed the data, performed the computation work, prepared figures and/or tables, authored or reviewed drafts of the article, and approved the final draft.
- Moldir Sagynay conceived and designed the experiments, performed the experiments, performed the computation work, prepared figures and/or tables, authored or reviewed drafts of the article, and approved the final draft.
- Shynar Mussiraliyeva conceived and designed the experiments, analyzed the data, performed the computation work, prepared figures and/or tables, authored or reviewed drafts of the article, and approved the final draft.
- Zhastay Yeltay conceived and designed the experiments, performed the experiments, performed the computation work, prepared figures and/or tables, authored or reviewed drafts of the article, and approved the final draft.

## Data Availability

The data and code are available in the Supplemental Files.

## Supplemental Information

Supplemental information for this article can be found online at http://dx.doi.org/10.7717/peerj-cs.3027#supplemental-information.

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
