# Peer review of "Detection of offensive content in the Kazakh language using machine learning and deep learning approaches"

_PeerJ Computer Science, doi:10.7717/peerj-cs.3027_

## Round 0.1 · original submission · Major Revisions

· Academic Editor

Major Revisions

Dear Authors,

Thank you for submitting your manuscript entitled "Detecting destructive content on social media using machine learning techniques" to PeerJ Computer Science. After reviewing the detailed reports from two referees, we have reached a decision of major revision.

The reviewers acknowledge the importance of your work, particularly its focus on destructive content detection in the Kazakh language. However, they raise several substantive concerns regarding the clarity of objectives, the methodological consistency, and the presentation and interpretation of results.
A key issue requiring clarification is the originality and provenance of the dataset. It is not fully clear whether the dataset used in the study is new or was previously published by the same authors. This point must be explicitly addressed. In addition, more detail is needed on the data collection and annotation process, including the number and background of annotators, annotation guidelines, inter-annotator agreement metrics, and any ethical protocols followed. These aspects are essential to assess the reliability and reproducibility of your findings.

Beyond the dataset, the manuscript would benefit from a clearer articulation of its contributions. The novelty of the proposed BERT + LSTM architecture needs to be better supported in relation to existing models. The related work section should be revised to provide a more critical and comparative view, particularly with respect to language-specific approaches.

Several methodological choices require stronger justification, including feature selection techniques, model selection, and the exclusion of multilingual transformer baselines such as mBERT or XLM-R. The results should be presented more clearly, with improved figures and tables, more comprehensive evaluation metrics, and, where possible, statistical validation or ablation analysis.

Finally, the manuscript should be carefully revised for grammar, formatting, and adherence to the journal’s style and sectioning requirements.

We encourage you to address these concerns in a revised version and submit a detailed, point-by-point response to the reviewers.

**Language Note:** The Academic Editor has identified that the English language must be improved. PeerJ can provide language editing services - please contact us at [email protected] for pricing (be sure to provide your manuscript number and title). Alternatively, you should make your own arrangements to improve the language quality and provide details in your response letter. – PeerJ Staff

Reviewer 1 ·

Basic reporting

This paper focuses on an important and timely topic: the use of machine learning approaches to identify destructive content on social media in the Kazakh language. I believe the title will be clearer if the language focus is reflected in it.

Overall, the introduction highlights the importance of this work. However, I suggest a better construction and organisation of objectives, aligning with the proposed methodology. For example, objective three mentions creating a model, even though this work suggests comparing multiple models.

The related work section discusses many previous works. However, overall, it lacks critical analysis between different works and details on how the reported previous work supports or motivates this study. Each paper is individually described as a separate item. Also, most of their language coverage is not mentioned; thus, it is unclear whether they focused on English, Kazakh or some other language. Moreover, the last paragraph, which states, “This literature review highlights the importance of developing language-specific models…” doesn’t seem to match the content in the related work, as most of the methods were referred to in using general terms like machine learning techniques, NLP algorithms, etc with no detailed discussion on language-specific models.

This paper also includes language and formatting errors, which need to be fixed by carefully proofreading. For example, line 33 repeats anonymously twice. Line 162 “Using automated text analysis, we extracted…”, “we” needs to be replaced by “they”. The citations which should be included within the parathesis are included in the text (e.g. line 31). Also, there are several single-sentence paragraphs (e.g. line 38, line 41), which need to be carefully organised. Also, please double-check the section numbers with the journal format, as numbers like 0.1, 0.2 seem incorrect.

Experimental design

Overall, I found it hard to follow the given model design. Section 0.7 Feature Selection on Page 8 mentions text data is converted into numerical vectors using a combination of TF-IDF and n-gram vectors. Following this, Section 0.8 Algorithms on Page 9 mentions the machine learning models used. Then, Section 0.9 describes a proposed model, which leads the reader into some confusion. Also, this section introduces a new embedding layer powered by BERT to convert text into numerical form without aligning with the previous content of the paper. I suggest clarifying this confusion, maintaining consistency among different sections and using clear section headings.

Also, this paper lacks justification for the model selection with supporting references. Are these models commonly used for this type of task?

Particularly, it is unclear why recent advanced multilingual transformer models such as m-BERT and XLM-R were not included in experiments using the transformer’s text classification architecture as they set the state-of-the-art performance for many languages.

It is also important to provide more information about the data annotation process, such as how many annotators were involved, their backgrounds, and inter-annotator agreements.

Validity of the findings

Overall, the presentation of the results needs to be improved. It is not very clear to refer to matrix representations in Figures 4(a) and 4(b), as their Y axis contains both categories and metrics together. Also, the image quality of given matrices is very low, making them harder to read.

The table references contain errors which need to be fixed. For example, line 598 states “…Table 3 demonstrate classification results in Bert algorithm.”, even though Table 3 contains results of both machine learning and deep learning methods.

The confusion in the model design also continues in the results section. Does the LSTM model follow the architecture given in Figure 3? If yes, how were the n-grams set for the results reported in Tables 3-8?

The models mentioned in Table 9 are not described in the methodology. Also, why is only accuracy reported for them?

Moreover, this paper lacks essential details about the experiments, such as training and testing data sizes, their statistics, and hyper-parameters used for different models.

Additional comments

Please refer to my comments above.

Reviewer 3 ·

Basic reporting

The manuscript is written in clear and professional English, but some parts could benefit from improved clarity, particularly in explaining the novelty of the proposed method.
Introduction and Background
The introduction and background provide context but lack a strong discussion on the research gap and how the proposed model overcomes limitations in existing methods.
The authors should explicitly highlight why their approach is necessary.
Literature references are present, but there is no clear connection between prior work and the proposed method. The authors should explicitly compare their model with previous approaches to emphasize improvements.
Figures and Data Presentation
Figure 1: It is unclear whether this represents an experimental setup or a general framework. If this is a guideline rather than an implemented methodology, it should be labeled accordingly.
Figure 3: The architecture diagram lacks details on hyperparameters and justifications for the chosen architecture. More explanation is needed on why LSTM is used instead of alternative sequence models.
Raw Data Policy
If experimental results are based on proprietary datasets, a statement should be included regarding data availability or alternative replication methods.

Dataset Collection Issues
The authors do not mention any guidelines or ethical considerations followed while collecting destructive speech data.
It is unclear how data was extracted—what keywords, hashtags, or linguistic patterns were used?
Authors should specify the number of annotators and their backgrounds to ensure annotation reliability. The paper does not mention which annotation tool was used.
There is no mention of how consistent the annotations were (e.g., Cohen’s Kappa, Fleiss’ Kappa, Krippendorff’s Alpha).

Experimental design

The research fits within the journal's scope, addressing text classification. However, the research question needs to be clearly defined in relation to the identified knowledge gap.
Methodology Concerns
The methodology lacks justification for key decisions:
Why were TF-IDF and stemming chosen instead of more advanced text processing techniques?
Why was an LSTM layer used after BERT embeddings instead of transformer-based sequence modeling or CNN-based encoders?
The experimental setup should specify dataset details, including data splits (train/test/validation), preprocessing steps, and the rationale behind model selection.
The baseline models should be explicitly compared to the proposed model to highlight improvements in performance.

Validity of the findings

The impact and novelty are not assessed clearly. The proposed architecture (BERT + LSTM) is widely used in NLP, and the authors should emphasize what makes their approach unique compared to existing work.
The evaluation metrics and dataset details should be more transparent to allow replication.
The results should include statistical analysis (e.g., confidence intervals, statistical significance tests) to validate model performance.
Robustness Tests
How does the model perform across different datasets?
Is the model overfitting to a specific dataset?
Are there ablation studies to show the contribution of each component (e.g., removing LSTM from the pipeline)?

Additional comments

The manuscript presents an interesting approach but requires more clarity on how it differs from existing architectures.
Figures should be revised for better clarity, including labels and explanations on why specific design choices were made.
A direct comparison between Figures 1 and 3 should be included to illustrate improvements and differences.
More discussion is needed on the practical implications and real-world applications of the proposed model.
No references to prior work supporting the chosen preprocessing steps, feature selection techniques, or model architecture.
Standard methods like BERT, LSTM, and embedding layers should be cited with relevant research papers.
The description appears to be general knowledge rather than being tailored to this study’s dataset or problem.
No justification for why these specific preprocessing steps, feature selection techniques, or model layers were chosen.
If they used pre-trained embeddings, which ones? If they trained embeddings from scratch, how were they trained?
Textual Issues
Lines 295-297: The text does not explain how the literature review was conducted (e.g., selection criteria, databases searched, keywords used).
Lines 204-212: The content discusses bot detection using machine learning models, but this does not establish a clear connection between bot detection and the main research goal.
Repetitive Phrasing:
Repeatedly using "This paper" and “This article” at the beginning of multiple paragraphs makes the writing repetitive and less engaging in the Related Work section.
Line 92-104: The phrase "for the classification of destructive web content in the Kazakh language on web resources" is repeated multiple times in different forms. Make the objectives more concise without repeating "Kazakh language" and "web resources" unnecessarily.
Line 75-78: The phrase "explore the state-of-the-art" suggests a comprehensive review of existing models, but the paper does not thoroughly analyze prior Kazakh-specific approaches.
Table Issues
Table 1: The dataset includes various categories such as violent, bullying, racism, nazism, and neutral. However, no justification is provided for the choice of these categories. Were they predefined, or did they emerge from data exploration?
Table 1: The dataset has a wide ID range (0 to 10189), suggesting a large dataset. However, class distribution is unclear—are all categories balanced? If not, is there a strategy to address imbalance?

---

## Round 0.2 · Minor Revisions

· Academic Editor

Minor Revisions

Thank you for the revised manuscript. The reviewer acknowledges clear improvements but raises a few issues that need to be addressed before acceptance:

Please clarify the rationale for using mBERT embeddings despite stating concerns about its limited Kazakh coverage, and discuss why KazBERT was not used.

The claim that the LSTM layer improves modeling of Kazakh complexity needs empirical or bibliographic support, or should be toned down.

Please explain the N-gram settings mentioned in Tables 4–9 to improve clarity.

We look forward to your revised submission.

Reviewer 1 ·

Basic reporting

The article has been improved significantly through the revisions.

Experimental design

There are still a few conflicts in the overall design. According to the further verifications provided in the revised version, the proposed approach aimed at language-specific modelling. The authors decided not to fine-tune multilingual transformers (i.e. mBERT, XLM-R) because they might lead to suboptimal performance due to the relatively small amount of Kazakh text in those models’ pretraining corpora. However, the reported experiments use mBERT embeddings with deep neural architectures, conflicting with the above idea on mBERT data coverage in Kazakh. Also, this conflicts with the overall aim of using language-specific models, as there is a BERT model tailored to Kazakh (KazBERT).

It is crucial to resolve these conflicts to clarify the proposed design and experiments.

Validity of the findings

The authors make a strong claim in the introduction “By integrating an LSTM layer after the BERT embeddings, the proposed model addresses the linguistic complexity of Kazakh more effectively than models relying solely on transformer outputs.” Since this is not covered in the experiments or supported by any references, its validity is in question.

Also, clarification of the N-gram setting in deep learning models, which is mentioned in Tables 4-9, is helpful for a better understanding of the reported findings.

---

## Round 0.3 · accepted · Accept

· Academic Editor

Accept

I consider that the paper is now suitable for publication.